# Tail assembly interference is a common strategy in bacterial antiviral defenses

Lingchen He [1,7], Laura Miguel-Romero[1,2,7], Jonasz B. Patkowski [1],
Nasser Alqurainy [3,4], Eduardo P. C. Rocha [5], Tiago R. D. Costa [1],
Alfred Fillol-Salom [1] ✉ & José R. Penadés [1,6] ✉

Many bacterial immune systems recognize phage structural components to activate antiviral responses, without inhibiting the function of the phage component. These systems can be encoded in specific chromosomal loci, known as defense islands, and in mobile genetic elements such as prophages and phage-inducible chromosomal islands (PICIs). Here, we identify a family of bacterial immune systems, named Tai (for 'tail assembly inhibition'), that is prevalent in PICIs, prophages and P4-like phage satellites. Tai systems protect their bacterial host population from other phages by blocking the tail assembly step, leading to the release of tailless phages incapable of infecting new hosts. To prevent autoimmunity, some Tai-positive phages have an associated counter-defense mechanism that is expressed during the phage lytic cycle and allows for tail formation. Interestingly, the Tai defense and counter-defense genes are organized in a non-contiguous operon, enabling their coordinated expression.

Bacteria encounter a multitude of mobile genetic elements (MGEs) in various environments[1,2]. These elements, such as bacteriophages (phages), phage satellites, or plasmids, play a crucial role in bacterial evolution and pathogenesis by facilitating horizontal gene transfer between cells[3–5]. However, they can also impose a cost on recipient cells, as observed with lytic phages, which lead to the lysis of the infected cells[6]. To survive such infections and control horizontal gene transfer, bacteria have developed a diverse range of immune systems[7]. These systems are either carried by MGEs themselves or clustered in specific chromosomal locations known as defense islands[8–13].

To prevent MGE mobility, different immune systems have evolved to target specific stages of the horizontal gene transfer process (recently reviewed in ref. 7,14). One type, known as superinfection exclusion systems, acts at the membrane level by disrupting MGE adsorption or blocking dsDNA injection[15,16]. Other systems, such as CRISPR-Cas or restriction-modification systems, recognize invading dsDNA and degrade their nucleic acids in a site-specific manner[17,18]. In addition, some strategies involve interfering with essential processes such as host takeover mechanisms, DNA replication, or transcription[19–21]. Moreover, several systems function as abortive infections (Abi), inducing growth dormancy or cell death to prevent the release of new MGE particles[22,23].

Interestingly, since the immune systems carried by MGEs could somehow affect their own transfer, MGEs should encode counter-defense systems clustered within their genomes to circumvent the cognate immune strategies. Despite the abundance of immune systems in MGEs[24,25], only a few examples of MGEs encoding an immune system and its cognate counter-defense system have been reported[21,26–30].

Among MGEs, phage-inducible chromosomal islands (PICIs) are prevalent, small (~10–15 kb) chromosomally integrated elements found in more than 200 species, with many strains containing multiple

---

[1]Centre for Bacterial Resistance Biology, Imperial College London, London, UK. [2]Instituto de Biomedicina de Valencia (IBV), CSIC, Valencia, Spain. [3]Institute of Infection, Immunity and Inflammation, College of Medical, Veterinary and Life Sciences, University of Glasgow, Glasgow, UK. [4]Department of Basic Science, College of Science and Health Professions, King Saud bin Abdulaziz University for Health Sciences & King Abdullah International Medical Research Centre, Riyadh, Saudi Arabia. [5]Institut Pasteur, Université de Paris Cité, CNRS, UMR3525, Microbial Evolutionary Genomics, Paris, France. [6]School of Health Sciences, Universidad CEU Cardenal Herrera, CEU Universities, Alfara del Patriarca, Spain. [7]These authors contributed equally: Lingchen He, Laura Miguel-Romero. ✉e-mail: a.fillol-salom@imperial.ac.uk; j.penades@imperial.ac.uk

of these elements[31–33]. PICIs are closely associated with certain temperate (helper) phages, parasitizing their life cycles[34,35]. In the presence of active helper phages, PICIs replicate extensively, excise from the bacterial chromosome, and are efficiently packaged into infectious particles composed of either helper phage or PICI virion proteins[36–38], facilitating intra- and inter-species transfer of these genetic elements[39,40]. Due to their ability to carry and disseminate an impressive array of immune systems[12,41], we initiated this study to characterize some of these systems.

Here we have identified a unique family of immune systems prevalent not only in PICIs but also in other families of MGEs, including prophages and P4-like phage satellites, among *Proteobacteria*. Members of this family of immune systems recognize and inhibit phage tail assembly formation of the infecting phage, resulting in tailless phages that are incapable of infecting new hosts. As the carriage and expression of this system may also affect phages encoding the immune system, we demonstrate the presence of a counter-defense mechanism to prevent autoimmunity. Remarkably, the immune system and its counter-defense system are always associated in the phage genome as part of a noncontiguous operon. This genetic structure allows for the coordinated expression of the immune and counter-defense genes. Our study reveals that MGEs, such as prophages, protect their hosts by targeting a key stage of phage infection—tail formation—while encoding a counter-defense system that is only activated after induction of the resident prophage.

## Results

### Phage satellites encode multiple hot spots of immune regions

We previously demonstrated that PICIs can encode immune systems at either the 5' or 3' ends of these elements[12]. However, we realized that some of these elements may preserve additional localizations in their genomes for the carriage of accessory regions, including immune systems. Specifically, we observed the existence of one of these new regions in the middle of the PICI genomes, adjacent to the PICI *pri* gene (Fig. 1a). To test this hypothesis, we cloned two genes, from two different *Pectobacterium* PICIs, under the control of the arabinose-inducible promoter ($P_{BAD}$) and evaluated their anti-phage activity against phages from various species (our collection of *E. coli* and *Salmonella*), with different life cycles (temperate or lytic phages), and representing major phage families (Supplementary Table S1). Note that the proteins encoded by these two genes share 64% sequence identity (Supplementary Table S2). We selected these genes due to their lack of homology to previously known anti-phage systems, as neither DefenseFinder[42] nor PADLOC[43] identified them as part of the immune system repertoire.

Supporting our hypothesis, both genes conferred resistance against multiple phages (Fig. 1b). Although the phages that are inhibited by the *Pectobacterium* PICI-encoded immune systems are lambdoid phages, some of these phages share low homology in DNA sequence. Our findings represent the discovery of a unique immune system, which we have named the tail assembly inhibition (Tai) system (see below for details). Furthermore, we have identified a new immune region within the PICI genomes.

### Tai immune systems are widespread in nature, encoded by diverse mobile genetic elements

After establishing the role of Tai genes in immunity, we decided to examine whether this immune system was widespread in bacteria. Initially, we searched specifically for homologs of *tai*-like genes in the NCBI database and compiled a dataset of 57 non-redundant homologs identified in the genomes of the *Proteobacteria* clade (Fig. 1c and Supplementary Table S3). These homologs were found in 17 species belonging to different bacterial phyla such as β- and γ-*Proteobacteria* (Supplementary Table S3). To analyze their distribution, we constructed a phylogenetic tree based on the 57 homologs and found that

they grouped into multiple clades without clear separation between their species origin (Fig. 1c).

Next, we sought to determine the genomic context carrying these genes. Notably, we found that most of these homologs were carried by phages (moron loci), while just a small subset was encoded by phage satellites, specifically P4-like elements and PICIs (Fig. 1c, Supplementary Fig. S1, and Supplementary Table S3). Within the phages, the genes encoding Tai-like homologs were predominantly located upstream of the packaging module, as observed in the prototypical *E. coli* phi80 (Supplementary Fig. S1a). However, in certain cases, they were also found in other recognized hotspots of immunity[44,45] or even in novel hotspots, as diagrammed in Supplementary Fig. S1a. Regarding the P4-like satellites, we observed that the *tai* genes were positioned in the recently identified hotspot near the *cos* site region[25], while in the PICIs, these genes were carried in the new accessory region localized downstream of the *pri* gene (Supplementary Fig. S1b and Fig. 1). These observations confirm a widespread distribution of Tai-like immune systems across different hotspots of immunity in different MGEs.

### Tai-like homologs also block phage reproduction

We next tested whether other Tai variants, located within phages, could mediate anti-phage immunity. We analyzed the phage variants since Tai systems are predominantly present in these MGEs. We named the different systems based on their species of origin and the specific version tested. Thus, TaiEcA means it is from *E. coli* origin, version A. We selected a total of 10 distinct Tai-like homologs from *E. coli* and *Salmonella* phages. These homologs exhibited low sequence identity, typically ranging from 22% to 43% amino acid identity (Supplementary Table S2). The genes were cloned on plasmids under the control of the $P_{BAD}$ promoter, introduced into the non-lysogenic strains 594 (for *E. coli*) and LT2 (for *Salmonella enterica*), and the strains carrying the different plasmids were subsequently challenged with the same array of phages used previously. Most of the Tai-like homologs tested clearly conferred immunity against phages, with only two versions (TaiSalE and TaiSalG) showing no detectable immune activity against the tested phages (Fig. 2a). We cannot exclude that they confer immunity against other phages not included in our panel. Importantly, while most of the tested versions targeted the same phages, the TaiEcC version affected different phages compared to the other versions (Fig. 2a). In a few cases, we observed that only the plaque size was affected, while in most of the cases, the phage titers were severely reduced (Fig. 2a). The fact that the Tai systems inhibit phages which have low homology in DNA sequence could be indicative that the different homologs may recognize a conserved phage process or detect molecular patterns that are present in different phages, as demonstrated previously with other immune systems[46,47].

### Tai systems protect the bacterial community of phage depredation

To test whether the Tai systems act at the cellular or population levels, we infected the recipient strains expressing the different Tai systems in liquid media with phages HK544 or HK578, and measured cell survival after phage infection. In these experiments, we used different multiplicities of infection (MOI; 0.1, 1, and 10). While the strain containing the empty plasmid completely lysed after infection, the strains that expressed the Tai-like homologs did not reveal significant lysis when the cells were infected at an MOI of 0.1. However, when higher MOI was used, these strains were partially (MOI of 1) or completely lysed (MOI of 10) after infection (Supplementary Fig. S2a, S2b). This indicates that although at higher MOI the strain always collapsed, at lower MOI (of 1) the different Tai versions showed different degree of protection.

Next, we decided to test the population mechanism of defense in a natural scenario, and the TaiEcA version was selected in these experiments for two reasons: (i) TaiEcA is encoded by the prototypical *E. coli* phi80 prophage, and (ii) a lysogenic strain carrying phi80

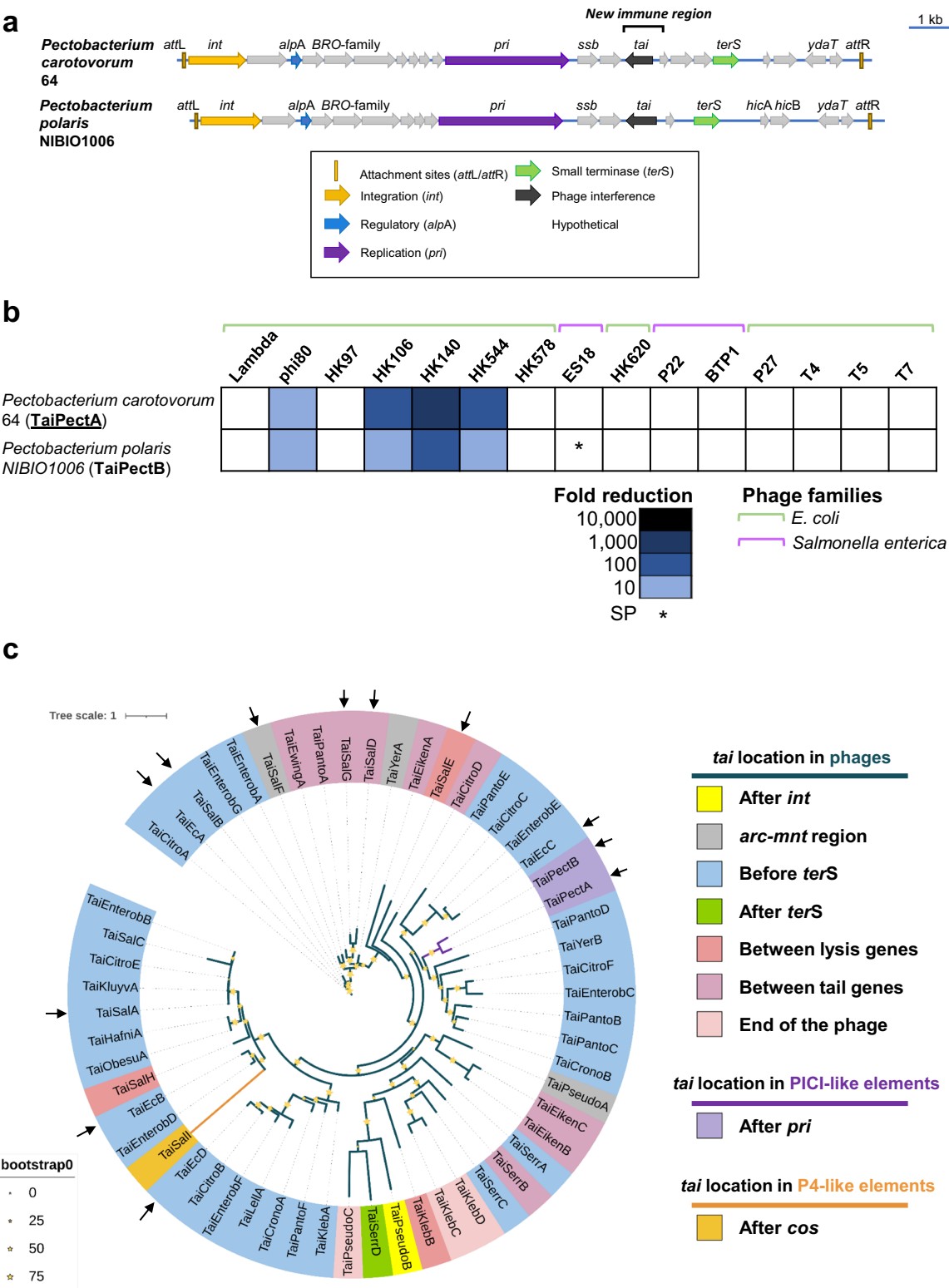

**Fig. 1 | PICIs contain a new accessory region with anti-phage defense systems.**
**a** PICI genomes are aligned according to the PICI convention, with the integrase gene (*int*) at the left end. Genes are colored according to their sequence and function: *int* is yellow; transcription regulator is blue; replication gene is purple; packaging genes are light green; genes encoding putative phage resistance proteins are black; genes encoding hypothetical proteins are gray. **b** Putative PICI-immune systems were tested against *E. coli* and *S. enterica* phages. Heatmap represents the phage fold-change protection, which was measured using serial dilution spot assay plaque. To calculate the fold-change reduction, it was compared

the efficiency of phage's plating on strains carrying either the empty plasmid or the plasmid expressing the immune system under study. The data is representative of three replicates. SP represents a small plaque phenotype. **c** Phylogenetic tree analysis of Tai-homologs found in phage (dark green branch), PICIs (purple branch), and P4-like element genomes (orange branch). The ring represents the location of *tai* systems in different MGEs. Bootstrap values are indicated. The phylogenetic tree was generated using a set of 57 non-redundant Tai sequences. Arrows highlight the Tai versions tested.

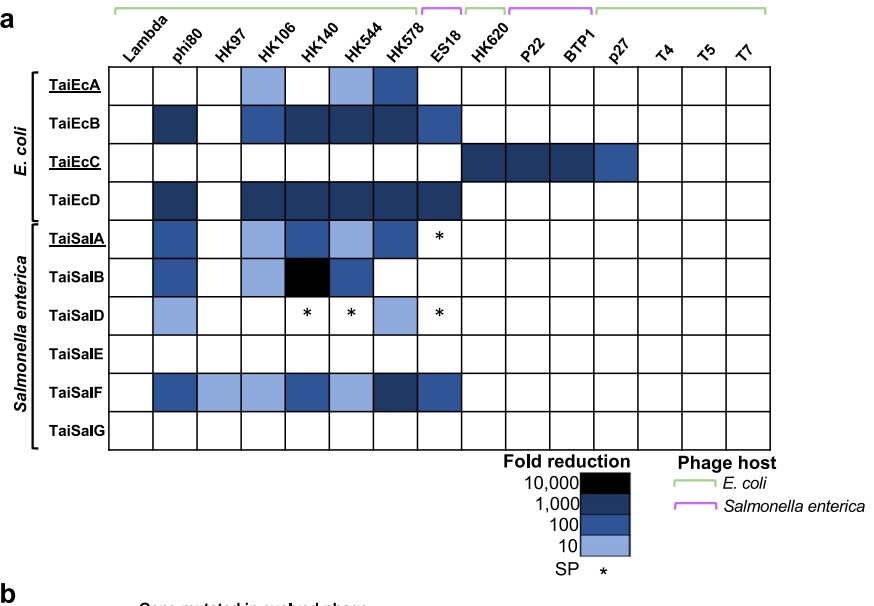

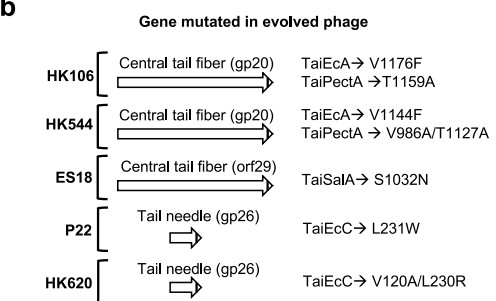

**Fig. 2 | Tai homologs are bona fide anti-phage systems. a** Tai immune versions from *E. coli* and *S. enterica* prophages were challenged against multiple phages. Heatmap represents the phage fold-change protection, which was measured using serial dilution spot assay plaque. To calculate the fold-change reduction, it was compared the efficiency of phage's plating on strains carrying either the empty plasmid or the plasmid expressing the immune system tested. The data is representative of three replicates. SP represents a small plaque phenotype. **b** Localization of the mutations in the HK106, HK544, ES18, P22 and HK620 evolved phages.

inhibits phage HK544 and HK578 reproduction dramatically, a process that depends on the presence of the phi80-encoded *tai* gene (Fig. 3a). When the lysogenic strain for phi80 was used, similar results to those in which TaiEcA was overexpressed from a plasmid were obtained (Fig. 3b and Supplementary Fig. S2c). Then, we measured the number of infecting particles generated after infection at different MOIs. Independently of the MOI used, the number of infecting particles generated after these infections was severely reduced (Fig. 3c and Supplementary Fig. S2d). Similar results were obtained when we performed a one-growth experiment to assess the defect caused by the phage progeny by TaiEcA (Fig. 3d). These results suggest that the Tai immune system acts as a population mechanism of defense.

**Characterization of evolved phages insensitive to Tai systems**
To dissect Tai's function, we conducted evolutionary experiments involving different phages and Tai systems (Supplementary Table S2). Despite our attempts to evolve phages against all immune versions, we only succeeded in evolving phages HK106 and HK544 against TaiEcA and TaiPectA, phage ES18 against TaiSalA, and phages P22 and HK620 against TaiEcC. Remarkably, all the evolved phages uniquely presented one or two amino acid substitutions (depending on the phage) in the tail proteins encoded by these phages (Fig. 2b). In the case of phages that encode the central tail fiber (also known as the tip attachment protein J), the mutations were consistently located in the C-terminal region of the central tail fiber. This protein corresponds to gp20 in phages HK106 and HK544, and gp29 in phage ES18. The central tail fiber is essential to produce the tail particle since it forms part of the tail tip complex assembly which is used to self-assemble the other tail components to produce the tail particle. Although with slightly different sequences in the J protein, the mutations observed in the evolved HK106 and HK544 phages were in the same residues. Another interesting phenomenon that occurred with the HK106 and HK544 phages was that when challenged against TaiEcA and TaiPectA, we obtained, in all cases evolved phages carrying mutations in the central tail fiber. However, these mutations were different, depending on the Tai, suggesting that although related, different Tai proteins may target the same proteins using different residues. In support of this, the evolved HK106 and HK544 phages were only insensitive to the inhibition of the *tai* immune system from they were selected (Supplementary Fig. S3). Finally, it is important to mention that the HK106 evolved phage against TaiPectA exhibited a noticeable fitness cost. Upon induction of the evolved phage, lysates were generated with 10x times fewer phages compared to the WT phage (Supplementary Fig. S3b). In addition, this HK106 evolved phage against TaiPectA formed smaller plaques than those observed for the WT phage (Supplementary Fig. S3b).

For the P22 and HK620 phages that evolved against TaiEcC, the mutation was found in the C-terminal region of the tail needle gp26 protein (Fig. 2b), which is responsible for plugging the DNA exit channel and penetrating the host cell envelope[48]. Since the tip attachment protein J plays crucial roles in both recognizing the receptor in the mature particle and initiating the tail phage formation[49], we hypothesized that the Tai proteins could act either as superinfection exclusion systems or blocking tail formation.

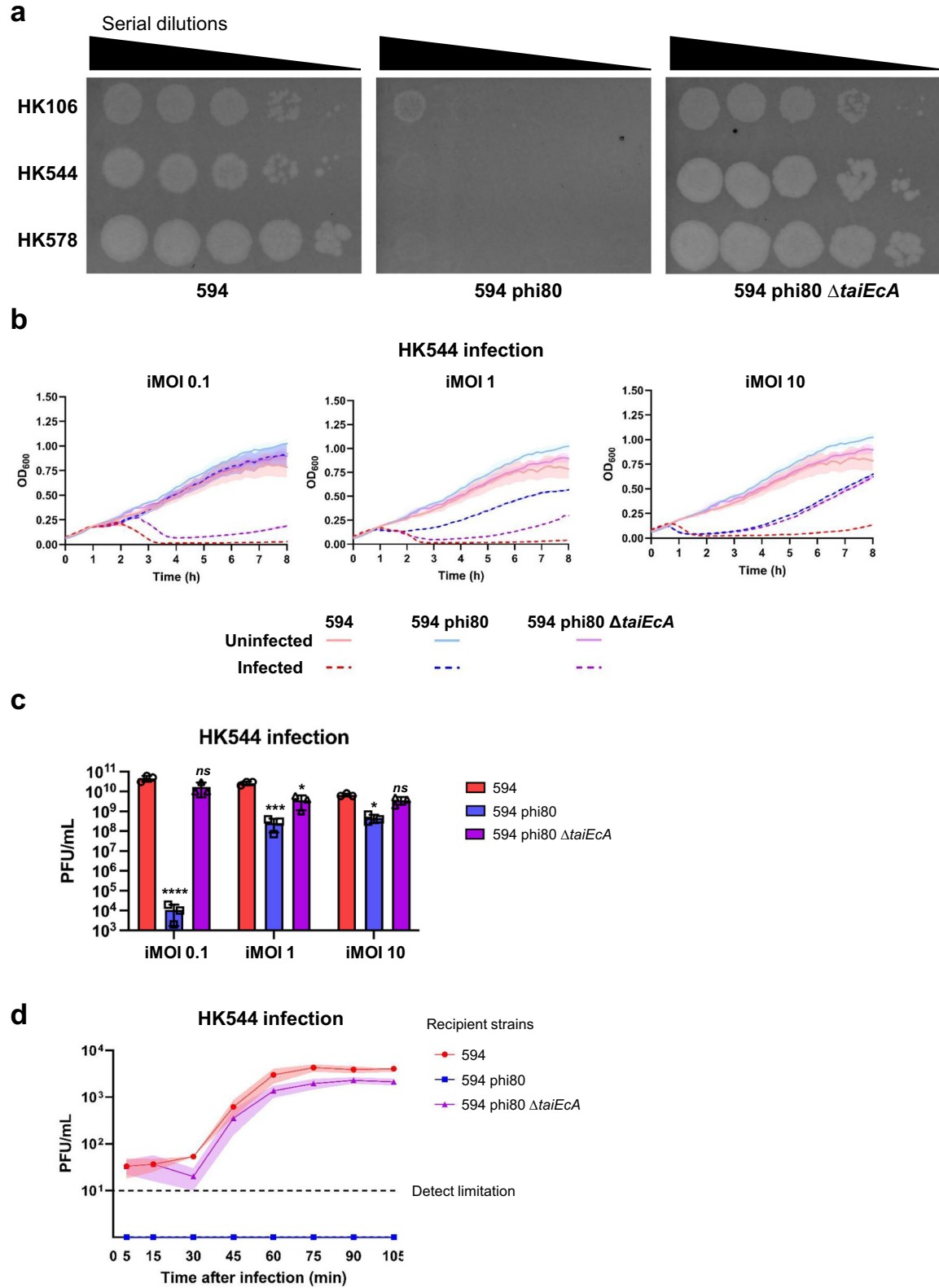

**Fig. 3 | *Escherichia coli* phi80 naturally protects the bacterial population.**
**a** Phages HK106, HK544, and HK578 were spotted on non-lysogenic *E. coli* 594 or its lysogenic derivatives carrying phi80 or phi80 Δ*taiEcA*. Ten-fold phage dilutions are shown. **b** Non-lysogenic or lysogenic strains for phi80 and phi80 Δ*taiEcA* were infected with HK544 phage at an input MOI (iMOI) of 0.1, 1, or 10. **c** Phage titers were determined from (**b**) using *E. coli* 594 as the recipient. A one-way ANOVA with Dunnett's multiple comparisons test was performed to compare results from phage infection in a non-lysogenic strain against lysogenic strains. Adjusted *p* values were

as follows: *ns* > 0.05; *\*p* ≤ 0.05; *\*\*p* ≤ 0.01; *\*\*\*p* ≤ 0.001; *\*\*\*\*p* ≤ 0.0001. The exact statistical values for each of the conditions tested are listed in Supplementary Table S4. **d** Non-lysogenic or lysogenic strains for phi80 and phi80 Δ*taiEcA* were infected with HK544 to perform a one-growth experiment. Samples were taken at different time points and phage progeny was measured. For (**b**), (**c**), and (**d**), the means and SD of three independent experiments are represented (*n* = 3). For (**a**), the experiment was repeated twice with similar results. Source data are provided as a Source Data file.

Superinfection exclusion systems operate at the membrane level, preventing the penetration of incoming dsDNA phage. To test whether the Tai proteins block phage injection, we performed Southern blot analyses to examine whether the phage HK106 could inject its DNA and replicate in the presence of the phage 80-encoded TaiEcA protein. We observed that phage HK106 replicated to the same extent in the presence or absence of TaiEcA (Supplementary Fig. S4a), ruling out the idea that TaiEcA functions as a superinfection exclusion protein. This was further supported by testing the HK106-mediated transfer of a plasmid carrying the HK106 *cos* site (pJP2786). The lysogenic strain containing phage HK106 and plasmid pJP2786 was mitomycin C (MC) induced, and the transfer of the phage and the plasmid was evaluated in the presence or absence of the phage 80-encoded TaiEcA protein. Consistent with the previous results, the presence of TaiEcA dramatically reduced the phage titer but did not affect the transfer of the pJP2786 plasmid, corroborating that TaiEcA does not act as a superinfection exclusion system (Supplementary Fig. S4b). Finally, to confirm that the other *E. coli* immune system versions are not superinfection exclusion systems, we tested the impact of different Tai systems, cloned on plasmids, on the induction of resident prophages. If the immune systems would function as superinfection exclusion proteins, we would expect the phage titers would not be affected. However, this was not the case, and the phage titers obtained after MC induction of the prophages in the presence of the Tai proteins were dramatically reduced (Supplementary Fig. S4c), confirming that they do not act as superinfection exclusion systems. Importantly, and using this approach, we observed that the TaiEcC system not only blocked infection of P22 phage but also induction of P22 and P27 (Fig. 2a and Supplementary Fig. S4d). This result highlights the ability of some Tai systems to block unrelated phages and together with the other systems, indicate that Tai systems have evolved to target most of the phage families.

## Tai systems inhibit the phage tail assembly pathway

Next, we investigated whether the immune systems inhibit tail formation. To test this, we purified HK106 and HK544 phage particles obtained in the presence or absence of different *E. coli* Tai proteins. Firstly, the different lysogens expressing the various Tai versions were treated with MC. Subsequently, the phages were precipitated with polyethylene glycol, purified by CsCl gradient, and visualized by electron microscopy. Consistent with tail formation inhibition, only empty capsids (tailless phages) were observed when the TaiEcA, TaiEcB, and TaiEcD versions were expressed during prophage induction, while complete phage particles were observed in the presence of the empty plasmid and TaiEcC version (Fig. 4a and Supplementary Table S4). To clearly confirm that the defect observed was due to the lack of tails and not as a consequence of the Tai proteins inhibiting the attachment of the formed tail to the mature capsids, we included the HK106 ΔterL phage in our experiments. This mutant phage does not produce the large terminase and, therefore, produces empty procapsids and tails but no functional particles. While the procapsids and tails were observed in the HK106 ΔterL polyethylene glycol preparation, only capsids were visualized in the presence of TaiEcA (Supplementary Fig. S5a), confirming that the TaiEcA protein blocks tail formation. Finally, the evolutionary experiments suggested that the Tai systems interact with the protein J, blocking its function. However, we weren't able to confirm this interaction using pull-down assays since the J protein is insoluble after expression.

Next, we decided to analyze the effect of TaiEcC on phage P27 progeny using the same methodology as for phages HK106 and HK544. Note that the P27 phage belongs to a different phage family (previously known as *Myoviridae*) and also assembles the tail and capsid independently. Consistent with tail formation inhibition, TaiEcC causes the formation of tailless particles (Fig. 4b). Furthermore, while we observe free tails on the sample with the empty plasmid,

the overexpression of TaiEcC abolishes the presence of free tails particles.

While the tail and head assemblies occur independently in HK106, HK544, and P27 phages, in phages P22 and HK620, the tail components assemble directly into the capsid particle. Three genes, *gp4*, *gp10*, and *gp26*, are required to stabilize the condensed DNA within the phage capsids. These proteins form the neck of mature virion particles, with Gp4 being the first one to be added and acting as an adapter for Gp10 and Gp26. Therefore, we subjected purified particles from P22 to electron microscopy in the presence or absence of TaiEcC. Although the tail is short, we observed that in the absence of TaiEcC, the P22 virion particles were filled with dsDNA (Supplementary Fig. S5b). However, in the presence of TaiEcC, the virion particles were empty (Supplementary Fig. S5b). This result suggests that the immune system prevents Gp26 from interacting with the mature capsid, resulting in virion particles that are unable to penetrate the host outer membrane. In addition, due to the absence of Gp26, the dsDNA encapsulated into the virion particles becomes loose, as suggested by electron microscopy. Thus, similar to the Tai immune systems that produce tailless phages, phage particles are produced but are unable to infect new hosts, protecting the population. Taken together, it is remarkable how two related proteins have evolved to target similar processes in unrelated phages. Although we suggest a direct inhibition of phage tail components via the Tai immune system, we cannot exclude that some host factors could also be involved in this process.

## *Escherichia coli* phi80 encodes a counter-defense system to prevent autoimmunity

Analyzing the pattern of inhibition by different immune systems (Fig. 2a), we made an intriguing observation. *E. coli* phi80 not only encodes TaiEcA but also the protein that is targeted and inhibited by this system. In fact, the C-terminal region of the central tail fiber protein encoded by phi80 is identical to the one found in phages HK106 and HK544, which are highly affected by TaiEcA (Supplementary Fig. S6a). This suggested that either our identification of the protein targeted by TaiEcA was incorrect, or phi80 could encode a counter-defense system to block at some point TaiEcA's activity. To test this, we generated different phi80 mutant prophages by deleting various non-essential regions from the phage genome. These mutant prophages were induced and challenged against the TaiEcA immune system. While the WT 80 phage is insensitive, we hypothesized that the mutant defective in the gene that encodes the counter-defense mechanism would be severely affected.

One of the new phi80 versions, carrying a deletion in the region encompassing genes *orf62* to *orf66* (phi80 Δ*orf62-orf66*), showed susceptibility against TaiEcA (Fig. 5a). Interestingly, the *taiEcA* gene (*orf63*) is encoded within that region. This led us to hypothesize that the gene encoding the counter-defense mechanism might be located adjacent to the immune system, similar to what has been observed in *Staphylococcus aureus* prophages and in *Salmonella* phage BTP1[26,50]. To test this, we individually deleted genes *orf62*, *orf64*, *orf65*, and *orf66*. Our hypothesis was that in the absence of the counter-defense, the phage reproduction would be severely affected in the presence of the immune system, resulting in a reduced titer. Supporting this idea, the titer obtained after induction of the phi80 Δ*orf64* prophage was significantly reduced (Fig. 5b). However, this defect disappeared when the mutant prophages were complemented in *trans* with a pBAD18 derivative plasmid expressing *orf64* (Fig. 5c). Based on its function (see below), the orf64 was renamed *atiEcA*, for anti-tail assembly inhibition *E. coli* version A.

Interestingly, despite the severe reduction in phage titer observed after induction of the phi80 Δ*atiEcA* mutant prophage, most of the plaques obtained with this mutant phage showed a normal size, which was unexpected. We postulated that these phages might have acquired additional mutations elsewhere in their genomes. To investigate this

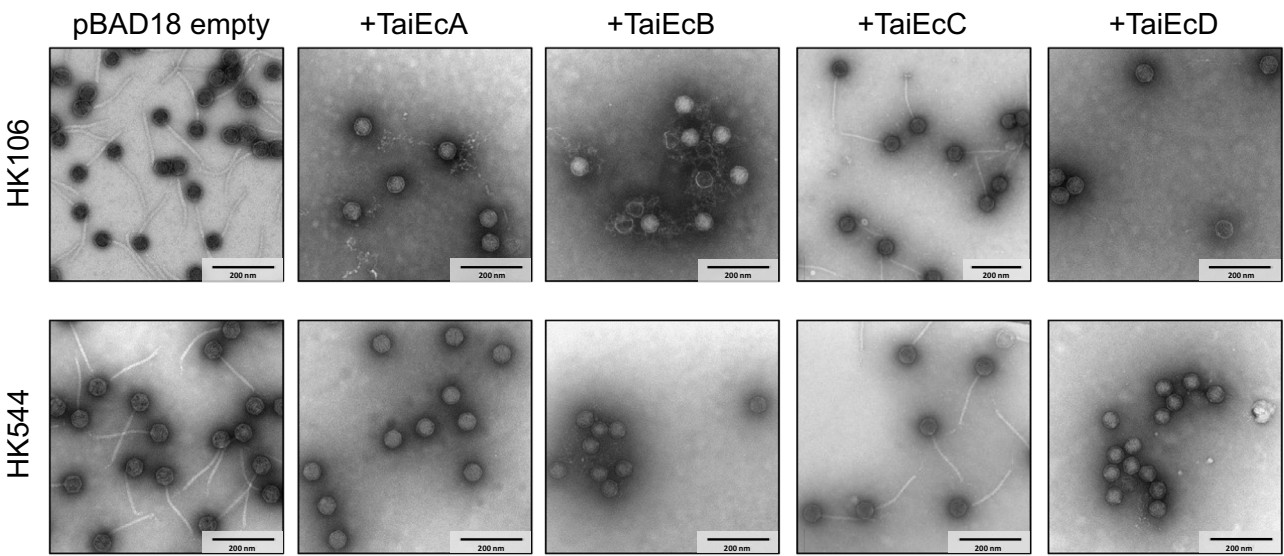

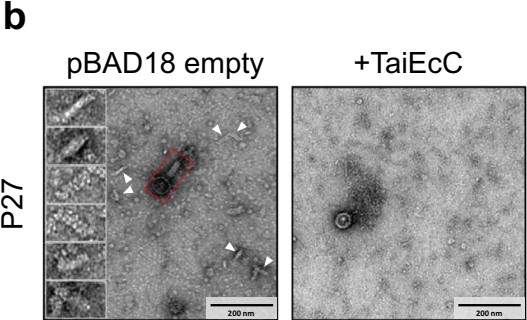

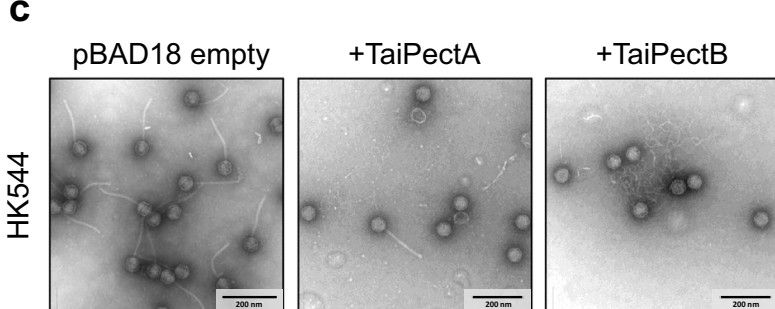

**Fig. 4 | Tai immune system inhibits the tail assembly.** Electron microscopy imaging of the lysates obtained after MC-induction of the HK106, HK544, or P27 lysogenic strains in the absence (pBAD18 empty) or presence of phage-encoded (**a**) and (**b**) or PICI-encoded (**c**) Tai immune systems. **a** Presence of either of the phage-encoded TaiEcABD proteins has a clear disruptive effect on phages HK106 and HK544, causing the formation of non-infective tailless capsids. **b** While TailEcC has no effect on HK106 or HK544, it displays specificity to P27 phage: in the absence of TailEcC, abundant free P27 tails were visible throughout imaging (white arrows and inset) together with fully assembled virions (red dashed frame); whereas the presence of TailEcC causes formation of tailless capsids and a striking lack of free tails. **c** The unique PICI-encoded immune systems are as effective as those encoded by phages, causing tail dislodgement. For (**a**), (**b**), and (**c**), different fields are shown, containing complete virion particles or tailless particles. Only one replicate was analyzed per experiment. Scale bars represent 200 nm. Source data are provided as a Source Data file.

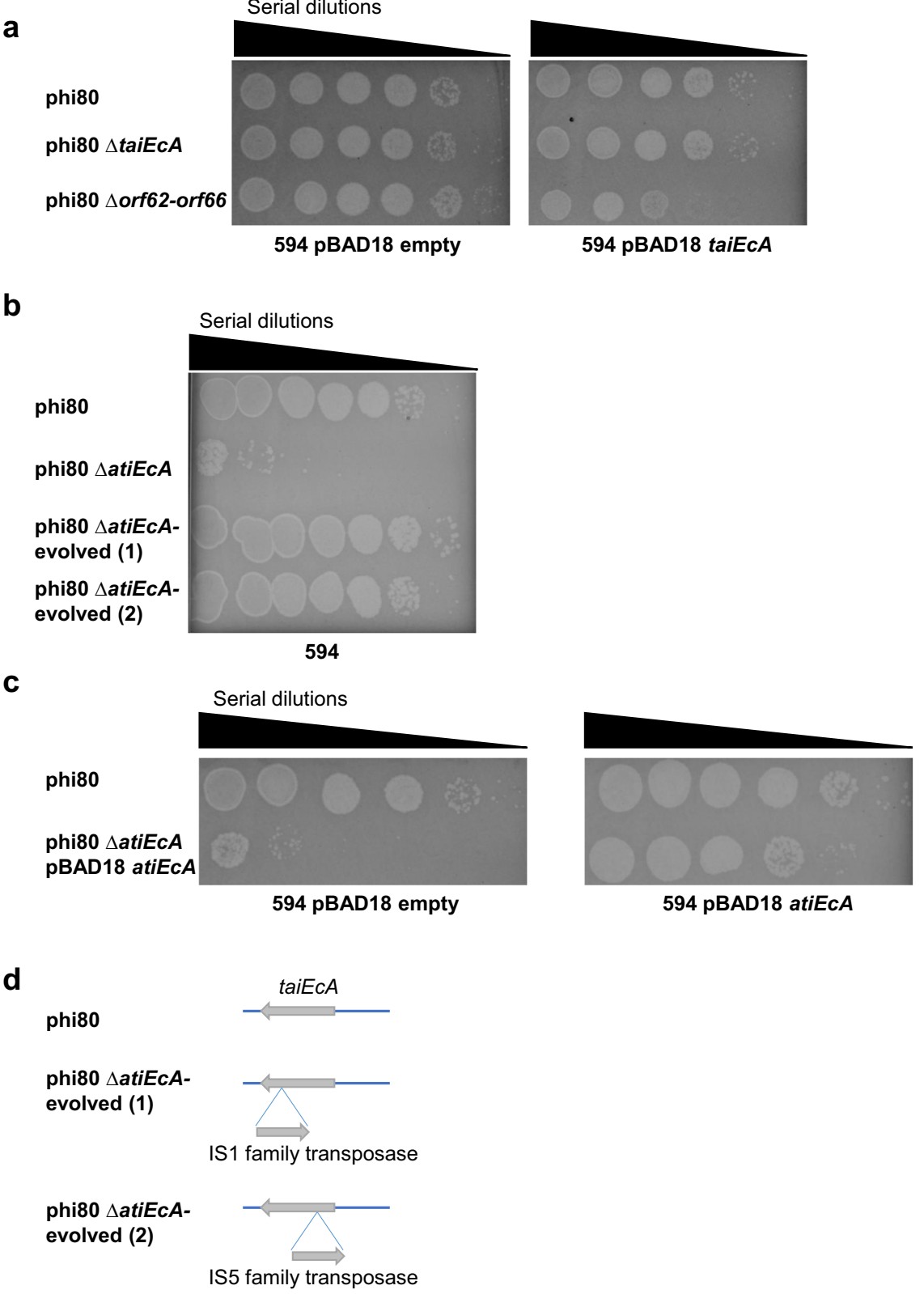

**Fig. 5 | *Escherichia coli* phi80 encodes a Tai immune system and its associated counter-defense mechanism. a** Phi80, phi80 Δ*taiEcA,* or phi80 Δ*orf62-orf66*, were spotted on non-lysogenic *E. coli* 594, containing empty plasmid or a plasmid expressing TaiEcA. Plates were supplemented with 0.02% arabinose. Ten-fold phage dilutions are shown. **b** Phi80, phi80 Δ*atiEcA,* or phi80 Δ*atiEcA* -evolved, were spotted on non-lysogenic *E. coli* 594. Ten-fold phage dilutions are shown. **c** Phi80 or phi80 Δ*atiEcA* complemented with pBAD18 *atiEcA*, were spotted on non-lysogenic *E. coli* 594, containing empty plasmid or a plasmid expressing *atiEcA*. Plates were supplemented with 0.02% arabinose. Ten-fold phage dilutions are shown.
**d** Schematic representation for transposon insertions into *taiEcA*. For (**a**), (**b**), and (**c**), the experiment was repeated twice with similar results. Source data are provided as a Source Data file.

further, we purified phages from normal plaques and obtained lysogens, which were then sequenced. Surprisingly, in all cases, we observed a transposase insertion into the *orf63* gene, disrupting the immune system gene (Fig. 5d). This indicates that, in the absence of the anti-antiphage system, the phages evolved mechanisms to inactivate their own immune system to avoid autoimmunity.

To confirm that *atiEcA* indeed encodes a protein that counteracts the activity of the TaiEcA system, we engineered a chimeric phage HK106 in which the *atiEcA* (*orf64*) gene was placed before the packaging module, similar to its arrangement in phi80 (Supplementary Fig. S1a). To maintain the size of the HK106 chimeric phage, the *atiEcA* gene replaced a gene that is not involved in the phage HK106 life cycle (see Supplementary Fig. S7a). The strains carrying either the WT or the chimeric HK106 prophages were induced, and the resulting phages were challenged against lysogenic strains carrying either the phi80 WT or the phi80 Δ*taiEcA* mutant prophages. As previously shown, the WT HK106 phage infected the lysogenic strain carrying phi80 Δ*taiEcA* normally but was blocked by the strain carrying the WT phi80 prophage. However, the chimeric phage expressing *atiEcA* infected both the WT and the phi80 Δ*taiEcA* lysogenic strains, confirming that the expression of the anti-antiphage system inhibits TaiEcA function (Supplementary Fig. S7b). Next, we searched the genomes used to identify the 57 non-redundant Tai homologs and whether they also encode an *atiEcA*-like gene (Supplementary Fig. S8a and Supplementary Table S5). We found a total of five *atiEcA*-like versions, which were present only in prophages. For instance, the lysogenic *Yersinia frederiksenii* strain *4821_8#8* encodes a TaiYerA immune system, which is 47% identical to TaiEcA. Interestingly, this prophage also encodes a counter-defense system that shares 40% protein identity with AtiEcA (Supplementary Fig. S8a). Although we only identified 5 examples of immune and counter-defense, we cannot exclude that the other immune systems encode a counter-defense element that does not resemble AtiEcA.

Finally, we tested whether the counter-defense could bind to the immune gene, preventing the immune function. We observed protein-protein interaction between the TaiSalB and AtiSalB versions. Thus, we hypothesize that each anti-Tai will bind to its cognate Tai version to inhibit its function (Supplementary Fig. S8b).

## The presence of *taiEcA* (*orf63*) and *atiEcA* (*orf64*) in a non-contiguous operon allows their coordinated expression

Our previous observations revealed that phi80 mediates phage HK106 interference through TaiEcA (Orf63) expression (Fig. 3a). This suggests that the *taiEcA* gene is expressed constitutively during the lysogenic state. In contrast, one would expect *taiEcA* not to be expressed during the lytic cycle, and/or *atiEcA* to be expressed after prophage induction to counteract TaiEcA's function. Notably, the genetic architecture carrying the phi80 *taiEcA* immune system and its associated counter-defense gene resembles the recently described noncontiguous operon[51], where the *taiEcA* gene is localized in a divergent orientation, interrupting the operon transcribed from Q, which contains, among other genes, *atiEcA* (Fig. 6a). Consequently, our model suggests that *taiEcA* is transcribed in the opposite direction to the rest of the phage late genes present in the operon controlled by Q (Fig. 6a). The Q protein is expressed only during the phage lytic cycle as Q is required to allow the production of functional virion particles. Q controls the expression of the phage late genes, which are involved in producing the capsid and tail components[52,53]. We hypothesize that this genetic arrangement will be crucial for switching off TaiEcA expression during the lytic cycle of the phi80 phage.

To verify if TaiEcA is expressed in the lysogen, we engineered a phi80 prophage capable of expressing a 3x-flagged version of the TaiEcA protein. Simultaneously, we aimed to generate a phi80 prophage expressing a tagged version of AtiEcA (ORF64). However, the size and localization of AtiEcA in the phage genome posed

challenges for this approach. To overcome this issue, we analyzed the expression of another protein whose gene is located in the same operon as the gene encoding the AtiEcA protein. Phi80 and phage λ have identical packaging modules, and their expression is controlled by the Q protein[52]. In these phages the antiterminator Q protein interacts with a DNA sequence in the late gene promoter segment and with RNA polymerase subunit[53], enabling the downstream expression of genes, including *atiEcA* and *orf6*. As *orf6* is positioned at the beginning of the late operon, we also engineered the phi80 prophage to express not only a tagged version of TaiEcA but also ORF6. It is worth noting that in this experiment, *the orf6* expression will mimic the *atiEcA* (*orf64*) expression.

Next, we MC induced the lysogenic strain for phi80, carrying the modified 3x-flagged *orf6* and *taiEcA*, and analyzed samples taken at 0, 60, 90, 120, and 150 min after MC induction using western blotting. As expected, Orf6 was only produced late during the lytic cycle of the phage (Fig. 6b). In contrast, the expression of the TaiEcA immune system occurred during the prophage state (Fig. 6B), and its expression continued at the same level after MC induction of the prophage, suggesting that the immune system is constitutively expressed.

To confirm that the expression of Orf6 is controlled by the Q protein, we introduced the pBAD18 plasmid, either empty or encoding for Q, into the aforementioned lysogenic strain for phi80 (3x-flag-*taiEcA*/3x-flag-*orf6*) phage. The strains were then induced with arabinose (which induces Q expression), and samples were taken at different time points. Supporting the idea that *orf6* belongs to the late operon, Orf6 was only produced when Q was overexpressed. Importantly, in the presence of the Q protein, TaiEcA expression was reduced (Supplementary Fig. S7c), suggesting that the transcript of the late operon somehow influences TaiEcA expression. In summary, the noncontiguous operon coordinates the expression of the immune and counter-defense genes.

## Discussion

Prokaryotic immune systems are diverse, and recent studies have highlighted that they are encoded not just in the bacterial chromosomes but also in MGEs. While bacteria and MGEs employ these immune systems to protect themselves against MGE infections, the immune systems encoded within MGEs are also used for inter-MGEs competition[3,12]. What are the consequences of these conflicts and how they impact bacterial and MGEs evolution, is an interesting topic of research.

Our study reveals a unique family of immune systems, named Tai (tail assembly inhibition), which is distributed in different MGEs, including phages and satellites (P4-like elements and PICIs). The Tai systems are encoded in known phage and satellite hot spots of immunity, whereas for PICI satellites, they are encoded in a new accessory region encoding immune systems. Once a phage infects a host that encodes the Tai immune system, this system inhibits tail assembly formation, releasing tailless phages incapable of infecting new hosts and protecting the population from phage predation (Fig. 4).

Although we only characterized some Tai versions, our data suggests a common strategy for all of them by interfering with a key component of the tail assembly pathway. This convergent strategy is employed by different Tai immune versions, even though they share low identity in protein sequence. For HK106 and HK544 phages, a platform called the tail tip complex is used to self-assemble the tail tube and to attach fibers and receptor-binding proteins, producing the tail component[54]. Importantly, the tail tip complex assembly begins with the central tail fiber, and the C-terminal portion of it interacts with the receptor on host cells. Our data suggests that the Tai immune systems tested target the central tail fiber, interacting with it and preventing it from performing its function and producing the tail particle. Indeed, the AlphaFold models suggest that the amino acid

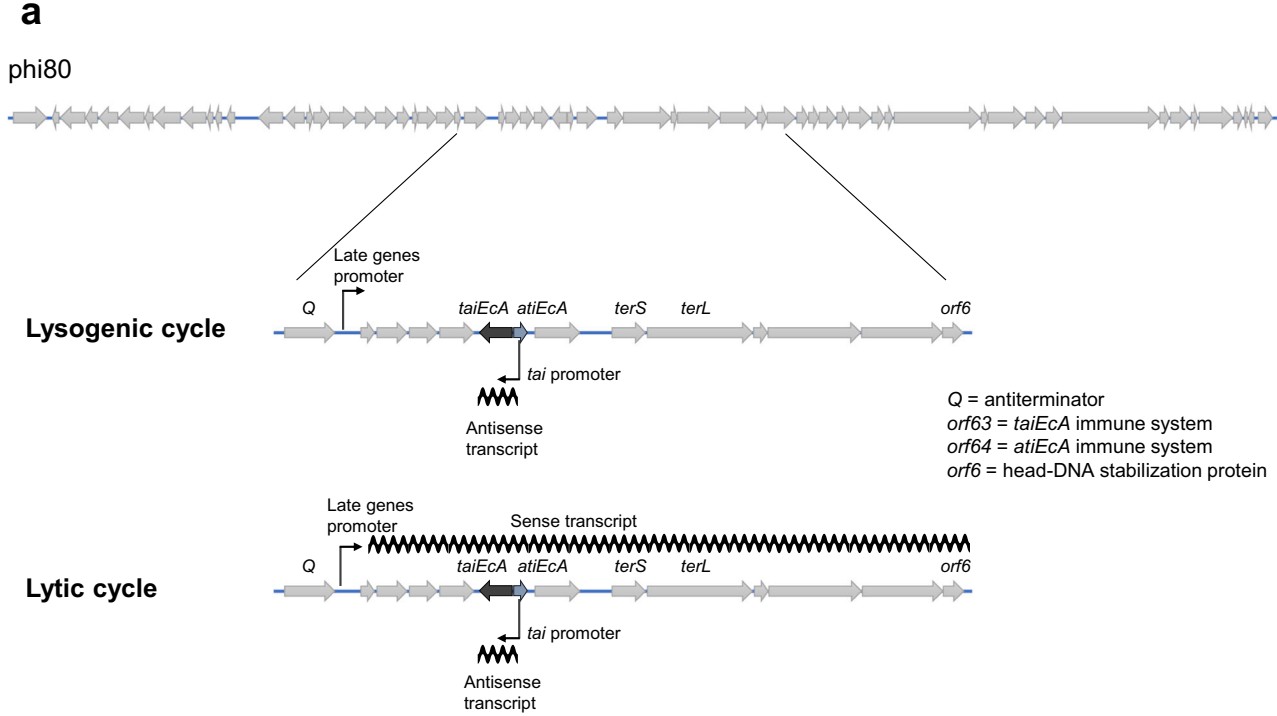

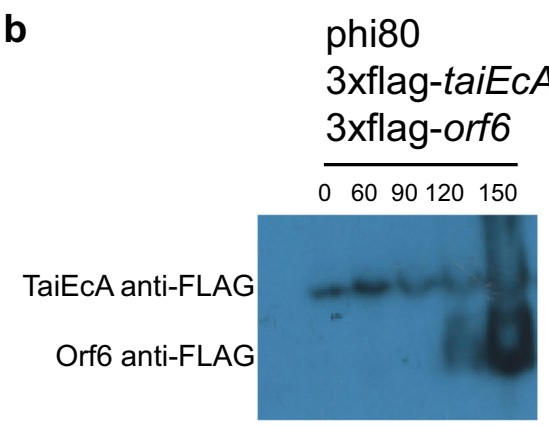

**Fig. 6 | A noncontiguos operon coordinates expression of the immune and counter-defense genes. a** Schematic representation of the phi80 genome, highlighting the region controlled by the Q antitermination protein and the localization of the *taiEcA*, *atiEcA*, *Q*, and *orf6* genes. The localization of the promoter controlling the late genes is also identified, as well as the directionality of the transcription. **b** Lysogenic strain phi80 3xflag-*taiEcA*/3xflag-*orf6* was MC-induced, and samples were harvested at 0, 60, 90, 120 and 150 min. A western blot using antibodies against the FLAG-tag carried by TaiEccA and Orf6 was performed. The protein molecular weight of TaiEcA is 26.1 kDa, while that of Orf6 is 14.3 kDa. The experiment was repeated twice with similar results. Source data are provided as a Source Data file.

that changes in the evolved phages (Supplementary Fig. S6b) is crucial for the oligomerization of the central tail fiber.

Strikingly, we observed that the TaiEcC homolog is able to target phages with different tail morphology (P22, HK620, and P27 phages). Although we only overcame immunity for P22 and HK620, we observed that the target is the tail needle protein. These phages directly assemble the tail components into the capsid particle, and the tail needle protein has a dual function: (i) plugging the dsDNA exit channel, and (ii) penetrating the host outer membrane. In this scenario, phages that lack the tail needle protein are incapable of infecting new hosts, and since the tail needle also plugs the dsDNA, it can be loosened as suggested by our data (Supplementary Fig. S5). Although we did not manage to obtain P27 phages insensitive to TaiEcC, our EM images suggest that this version recognizes a tail protein with a similar folding to the tail needle protein, interfering with the phage tail assembly process as well, a process that releases tailless phage particles.

While the tested Tai immune systems target lambdoid phages, it is surprising that the prototypical member of the lambdoid phages, λ, is not inhibited by any of the Tai immune systems tested so far. We could speculate that phage λ has evolved to encode a different J protein to avoid interaction with the various Tai immune systems. Similarly,

phage HK97, which has a longer C-terminal region on the central tail fiber gene (Supplementary Fig. S6), is only inhibited by one system. It is also tempting to speculate that this additional region is capable of evading immunity by most of the Tai immune versions.

Recent studies have identified several bacterial immune systems that sense phage structural components. Some systems, such as SEFIR[55], Lit[56], CBASS[57], Pycsar[58], PifA[59] or CapRel[46], recognize major capsid proteins. The Avs family immune systems sense the portal and terminase proteins[47], while other systems such as DSR2[60], Septu[55], or Tha[50] recognize tail components (the tail tube, the tail fiber, and the minor tail, respectively). Thus, the phage structural components are a recurring component recognized by different immune systems. However, the aforementioned immune systems act as abortive infection systems, sensing the phage infection though their structural component and activating their effector, which usually produces growth arrest or cell suicide[23]. In contrast, our work provides evidence that a phage structural pathway, specifically tail formation, is directly inhibited, producing non-viable tailless particles incapable of infecting new bacteria from the population. Notably, during the revision of this manuscript, similar mechanisms of interference were reported. In the first example, some *Pseudomonas aeruginosa* prophages were able to block the tail assembly of infecting phages by expressing a protein called Tab (from 'tail assembly blocker')[61]. In a more recent manuscript, a bacterial ISG15-like system from *Collimonas* sp. OK412 was able to block phage infection by releasing a mixture of partially assembled, tailless phage particles and assembled phages with non-functional central tail fiber. This occurs because this system obstructs the function of the central tail fiber by covalently attaching ubiquitin-like proteins to it[62]. In addition to structural genes, it is important to mention that other immune systems can be activated by other phage functions. For example, the Tin system encoded in P2 phages inhibits phage replication of T-seven phages[63], while the AbiZ system encoded in IS elements from a conjugative plasmid in *Lactococcus* causes premature lysis[64].

Immune systems recognize conserved pathways or components, and the presence of the different Tai immune system versions in phages suggests that those phages must have evolved to avoid autoimmunity by either adapting the conserved process/component or encoding a counter-defense system. Notably, despite the abundance of immune systems encoded by MGEs, only a few examples of associated counter-defense have been documented[21,26–30,50]. Here, we demonstrate that phi80, which encodes a Tai immune system, also encodes a counter-defense to avoid autoimmunity. Similarly, prophages expressing Tab are not inhibited during their own lytic cycle because they produce a counter-defense protein that neutralizes Tab's function[61]. Indeed, our data suggests that the phi80 counter-defense is specific to its own version (Fig. 2a); however, we cannot exclude the possibility of the existence of other versions whose role is to block the function of non-cognate Tai systems. In this scenario, one would expect that the tail proteins encoded by these phages do not interact with the cognate Tai protein. This idea can be important for the phage satellites since the expression of the Tai proteins could be detrimental to their own transfer. Thus, phage satellites encoding *tai* immune systems should target non-helper phages, encode a counter-defense mechanism, or employ a mechanism to switch off the immune expression once the satellite is activated.

Interestingly, for the Tai immune system described so far, the counter-defense is only produced during the phage lytic cycle. This behavior is due to the nature of the immune system target, as it is only produced in the lytic cycle. Therefore, the phage only requires activating its counter-defense once it enters the lytic cycle for its propagation. In addition, the immune and counter-defense genes are encoded together in a noncontiguous operon. This genetic structure allows the regulation of both genes, with the immune system constitutively expressed and the counter-defense expressed only when

required. Remarkably, the importance of noncontiguous operons in controlling the coordinated expression of the immune systems and their counter-defense mechanism has been unappreciated until recently. Recently, it has been shown that *S. aureus* prophages use this unique genetic structure to control the expression of immune systems and their connate anti-immune mechanisms[50]. Moreover, since phages recombine with other phages to produce new ones, the presence of immune and counter-defense genes together will favor their acquisition and avoid any possible detrimental effect for the new phage.

Because MGEs are a rich source of immune systems, we anticipate that novel immune systems blocking other structural component pathways will be discovered. This is an elegant strategy to block competing MGEs, as these unique systems will be able to target different phages at the same time with structurally related proteins. Moreover, to overcome autoimmunity, we predict that these new immune systems will encode their associated counter-defense, many of these pairs being encoded on noncontiguous operons. Finally, it has been recently proposed that both the mechanism that provides immunity, as well as those that prevent immunity, should be included as an intrinsic part of the phage life cycle[50]. Our results confirm this idea, as a defect in the immune system would make the lysogen more susceptible to infection, while a defect in the mechanism that prevents immunity would generate autoimmunity with the concomitant severe defect in the reproduction of the defective phage.

## Methods

### Bacterial strains and growth conditions

Bacterial strains, phages, plasmids, and oligonucleotides used are listed in Supplementary Tables S6, S1, S7, and S8, respectively. *E. coli* and *Salmonella enterica* strains were grown at 37 °C or 30 °C on Luria-Bertani (LB) agar or in LB broth with shaking (120 rpm). For antibiotic selection, ampicillin (100 μg ml⁻¹), kanamycin (30 μg ml⁻¹), or chloramphenicol (*cat*) (20 μg ml⁻¹), form Sigma-Aldrich, was added when required.

### Plasmid construction

The plasmids generated in this study (Supplementary Table S7) were constructed by cloning PCR products, amplified with the oligonucleotides listed in Supplementary Table S8, into the corresponding vector (pBAD18, pK03-Blue, pET28a, pUT18, and pKT25) using enzymatic digestion and ligation. Synthetic genes were purchased from ThermoFisher Scientific (Invitrogen GeneArt Gene Synthesis Services). Cloned plasmids were verified by Sanger sequencing (Eurofins Genomics).

### DNA methods

For *E. coli* phi80, gene deletions were performed as previously described[65], in which the *cat* resistance maker was amplified by PCR, with oligonucleotides listed in Supplementary Table S8, from plasmid pKD3 and inserted into the phage genome using λ Red recombinase-mediated recombination. Briefly, the PCR product was transformed into the recipient strain harboring plasmid pWRG99, which expresses the λ Red recombinase, and the markers were inserted into the phage genome. The different mutants obtained were verified by PCR. To eliminate the chromosomal marker from the phage genome, plasmid FLP helper pCP20 was transformed into the strains containing the resistance marker insertions. Strains harboring the plasmid pCP20 were grown overnight at 30 °C, and then, a 1:50 dilution (with fresh LB) was prepared and grown at 42 °C for 4 h to encourage plasmid loss while producing FLP recombination. Strains were plated out on LB plates, without antibiotics, and incubated at 37 °C for 24 h. Individual colonies were streaked out and PCR was performed to corroborate that the chromosomal marker had been removed. The different mutants obtained were subsequently verified by PCR and DNA sequencing (Sanger sequencing, Eurofins Genomics).

For *E. coli* HK106 chimeric phage, an allelic replacement strategy was performed[66]. Briefly, the allelic exchange vector, pK03-Blue, was used to clone the phage regions flanking the proposed *orf64* gene insertion using the oligonucleotides listed in Supplementary Table S8. Then, the plasmid was introduced into the strain carrying the prophage, and the transformants were selected on LBA plates supplemented with a *cat*. Plates were incubated at 32 °C for 24 h for the selection of the temperature-sensitive plasmid. To perform the homologous recombination, the plasmid was forced to recombine and integrate into the phage genome at the non-permissive temperature (42 °C). Then, light blue colonies (indicative of plasmid integration) were grown in LB broth at 32 °C and ten-fold serial dilution of the overnight cultures was plated on LBA supplemented with X-gal (5-bromo-4-chloro- 3-indolyl-B-D-galactopyranoside) and sucrose 5% (to force the plasmid loss). Plates were incubated at 32 °C for 24 h. White colonies (indicative of plasmid loss) were screened for chloramphenicol sensitivity. The HK106 chimeric phage obtained was verified by PCR and DNA sequencing (Sanger sequencing, Eurofins Genomics).

For 3xflag phage engineering, the site-directed scarless mutagenesis was performed to obtain the 3xflag-*orf63* (C-terminal) and 3xflag-*orf6* (N-terminal)[67]. Briefly, the *km*R marker, together with an I-*Sce*I recognition restriction site was amplified by PCR, using oligonucleotides listed in Supplementary Table S8. Next, the corresponding PCR product was electroporated into the recipient strain harboring plasmid pWRG99, which expresses the λ Red recombinase-mediated system used for homologous recombination. The *km*R marker + I-*Sce*I insertion was verified by PCR, and then 90mer DNA fragments derived from oligonucleotides, which contain the 3xflag, were electroporated into the mutant strain expressing the λ Red recombinase-mediated system. Successful recombinants were selected by expression of I-*Sce*I endonuclease. The different phi80 3xflag versions obtained were verified by PCR sequencing (Sanger sequencing, Eurofins Genomics).

## Phage plaque assays

For *E. coli* and *Salmonella enterica*, the corresponding strain carrying the pBAD18 empty or with immune systems was grown overnight. A 1:50 dilution (in fresh LB broth) of the overnight strains was grown until an $OD_{600} = 0.34$ was reached. Bacterial lawns were prepared by mixing 300 μL of cells with phage top agar (PTA) and poured onto square plates. Then, serial dilutions of phages were prepared in phage buffer (50 mM Tris pH 8, 1 mM $MgSO_4$, 4 mM $CaCl_2$, and 100 mM NaCl) and spotted on corresponding plates, which were supplemented with 0.02% arabinose to overexpress the immune system. Plates were incubated at 37 °C for 24 h. The fold change protection was measured as the number of plaques in the strain carrying the empty plasmid divided by the number of plaques in the strain carrying the immune system. Small plaques are represented as SP.

## Phage induction

For *E. coli*, overnight cultures of lysogenic strains carrying the corresponding plasmid were diluted 1:50 in fresh LB broth containing the corresponding antibiotic, and grown until an $OD_{600} = 0.2$. Prophages were induced by the addition of mitomycin C (2 mg ml⁻¹) and cultures were supplemented with 0.02% arabinose to overexpress the immune system. Following induction, cultures were incubated at 32 °C with slow shaking (80 rpm). Generally, cell lyses occurred after 5-6 h post-induction and the induced samples were filtered using sterile 0.2 μm filters (Minisart® single-use syringe filter unit, hydrophilic and non-pyrogenic Sartonium Stedim Biotech). The number of phage particles in the resultant lysate was quantified.

## Phage titration

For *E. coli* and *Salmonella enterica*, the corresponding strain carrying the pBAD18 empty or with immune systems was grown overnight. A 1:50 dilution (in fresh LB) of an overnight recipient strain carrying the empty plasmid or plasmids with the immune systems was grown until an $OD_{600} = 0.34$ was reached. Then, 50 μL of cells were infected with the addition of 100 μL of phage lysate serial dilutions prepared with phage buffer for 5 min at room temperature. Following incubation, 3 ml PTA was added to stop the infection process, and mixtures of culture-phage were plated out on phage base agar plates (PBA; 25 g of Nutrient Broth No. 2, Oxoid; 7 g agar) supplemented with $CaCl_2$ to a final concentration of 10 mM. Plates were incubated at 37 °C for 24 h. A Number of plaques formed (phage particles present in the lysate) were counted and represented as the plaque forming units (PFU/mL).

## Plasmid or phage transduction

For plasmid or phage transductions, a 1:50 dilution (in fresh LB) of an overnight recipient strain was grown until an $OD_{600} = 1.4$ was reached. A total of 1 mL of recipient cells, supplemented with 4.4 mM of $CaCl_2$, were infected for 30 mins at 37 °C with the addition of 100 μL of phage lysate diluted in phage buffer. Following incubation, 3 ml LB top-agar (LTA: 20 g LB, Sigma; 7.5 g agar, Formedium) was added to stop the transduction process, and the entire contents were poured onto LB agar plates containing the appropriate antibiotic. Plates were incubated at 37 °C for 24 h. A number of colonies formed (transduction particles present in the lysate) were counted and represented as the colony forming units (CFU/mL).

## Phage evolution

Phages were subjected to consecutive rounds of infections until isolated phage mutants could form clear plaques on strains expressing the corresponding immune systems. Note that the phage titer is reduced, and they produce small plaques on the different immune systems. The phage plaques obtained after infection versus an immune system were collected in a tube containing 3 mL of phage buffer. Afterward, tubes were centrifuged at 5000 × *g* for 5 min. The supernatant was filtered using a sterile 0.2 μm filter and the resultant lysate was used in a new round of phage infection. Consecutive rounds were performed until the phage overcame the immune system-mediated interference. We considered phage mutants when the plaques that they form are the phage WT infecting the strain without the immune system. Then, single plaques of phage mutants were selected and amplified, which were sequenced by whole genome sequencing by Microbes NG.

## Whole genome sequencing

Whole genome sequencing was performed at Microbes NG using Illumina, obtaining 2 × 250 bp pair-end reads, or at SeqCenter company (USA) using Illumina NextSeq 2000 platform, obtaining 2 × 151 bp pair-end reads. A minimum size of 200 Mbp sequencing package was obtained. Microbes NG performed a standard analysis pipeline, in which they identified the closest available reference genome using Kraken and mapped the reads to this reference genome using BWA mem to assess the quality of the data. Then, they performed a de novo assembly of the reads using SPAdes and mapped the reads back to the resultant contigs. Contigs obtained from Microbe NG analysis were compared to the phage WT genome to identify the nucleotide change. Trimmed reads from SeqCenter were mapped to the appropriate genome and were compared to the phage WT genome to identify the nucleotide changes.

## Growth curves

For *E. coli*, the corresponding strain carrying the pBAD18 empty or with immune systems was grown overnight. A 1:50 dilution (in fresh LB, supplemented with 0.02% arabinose) of the overnight recipient strains was grown until an $OD_{600} = 0.1$ was reached. Then, 200 μL of cells were seeded into a 96-well plate, and they were infected with the addition of phage lysate at a multiplicity of infection (MOI) of 0.1, 1, or 10 when appropriate. The plate was incubated in a FLUOstar Omega (BMG

LABTECH) plate reader at 500 rpm and 37 °C, with the $OD_{600}$ measured every ten minutes.

## One-step growth curve

Overnight cultures of recipient strains 594, 594 phi80, and 594 phi80 Δ*taiEcA*, were sub-cultured until CFU/mL = $10^7$. Phage HK544 was diluted to PFU/mL = $10^6$. Then, a total of 10 μL of HK544 was added to 990 μL of the corresponding recipient strain supplemented with 10 mM $CaCl_2$ (timepoint 0 min) and incubated at 37 °C. After 5 minutes, 1 mL of infected cells were centrifuged, and the supernatant was removed. After that, the pellet was resuspended with 1 mL of fresh LB. Then, 0.1 mL of resuspended pellet was added to 9.9 mL of pre-warmed LB following a 10-fold serial dilution with pre-warmed LB. All dilutions were incubated at 37 °C. Then, 100 μl of samples from each dilution were taken at defined time points and mixed with 10 ul of overnight culture of recipient strains respectively. A total of 3 mL of PTA was added to each mixture and plated to PBA plates. Plates were incubated at 37 °C for 24 h. A Number of forming plaques were counted and represented as plaque-forming units (PFU/mL).

## Southern blot analysis

Following phage infection, one milliliter of each sample was taken at defined time points and pelleted. Samples were frozen at − 20 °C until all time points were collected. Then, samples were re-suspended in 50 μL lysis buffer (47.5 μL TES-Sucrose and 2.5 μL lysozyme [10 μg mL$^{-1}$]; Sigma-Aldrich) and incubated at 37 °C for 1 h. Following incubation, 55 μL of SDS 2% proteinase K buffer (47.25 μL $H_2O$, 5.25 μL SDS 20%, 2.5 μL proteinase K [20 mg mL$^{-1}$], Sigma-Aldrich from *Tritirachium album*) was added to the obtained lysates and incubated at 55 °C for 30 min. Then, lysates were vortexed with 10 μL of 10x loading dye for 1 h. Following this incubation, samples were frozen and thawed in cycles of 5 min incubation in dry ice with ethanol and in a water bath at 65 °C. This was repeated three times. Chromosomal DNA was separated by agarose gel electrophoresis, samples were run on 0.7% agarose gel at 30 V, overnight. Then, the DNA was transferred to Nylon membranes (Hybond-N 0.45 mm pore size filters; Amersham Life Science) using standard methods. DNA was detected using a DIG-labeled probe (Digoxigenin-11-dUTP alkali-labile; Roche) and anti-DIG antibody (Anti-Digoxigenin-AP Fab fragments; Roche), before washing and visualization. The oligonucleotides used to obtain the DIG-labeled probe are listed in Supplementary Table S8.

## Electron microscopy

To analyze the phage particles by transmission electron microscopy (TEM), a large-scale induction was performed to precipitate and CsCl purify the phage particles. Briefly, a total of 100 mL of the corresponding lysate was produced by MC and arabinose induction. Then, lysates were filtered using sterile 0.2 μm filters and treated with RNase (1 μg mL$^{-1}$) and DNase (1 μg mL$^{-1}$) for 30 min at room temperature. Then, 1 M of NaCl was added to the lysate and incubated for 1 h on ice. After incubation, the mix was centrifuged at 11,000 × *g* for 10 min at 4 °C, and the supernatant was mixed with 10% wt/vol polyethylene glycol (PEG) 8000 and kept overnight at 4 °C. Following overnight incubation, phages were collected at 11,000 × *g* for 10 min at 4 °C and the pellet was resuspended in 1 mL of phage buffer. Only for Supplementary Fig. S5a, we analyzed the phage particles at this stage. Then, the precipitated phages were loaded on the CsCl step gradients (1.35, 1.5, and 1.7 g mL$^{-1}$ fractions) and centrifuged at 80,000 × *g* for 2 h at 4 °C. The phage band was extracted from the CsCl gradients using a 23-gauge needle and syringe. Phages were dialyzed overnight to remove CsCl excess using SnakeSkin™ Dialysis Tubing (3.5 K MWCO, 16 mm dry) into 50 mM of Tris pH 8 and 150 mM NaCl buffer. Then, 10 μL of the dialyzed samples were incubated in a carbon-coated copper grid for 5 min. Samples were fixed with 1% Paraformaldehyde for 2 min, before washing three times with distilled water for 30 sec. Afterward, the samples were stained with 2% Uranyl Formate for 30 sec, and they were allowed to dry at room temperature for 15 min. The Hitachi HT7800 or JEOL 1200 TEM microscope was used to examine the samples in the electron microscopy facility at Servicio Central de Soporte a la Investigación Experimental (SCSIE), University of Valencia, Valencia, or at the School of Life Sciences MVLS, University of Glasgow. Photos were obtained at 12 K,15 K, or 50 K of magnification.

The P27 phage was analyzed with slight changes to the procedure outlined above. Briefly, a total of 1 L lysate was produced by MC and 0.3% arabinose induction. The phage solution was cleared and precipitated as described above and solubilized in 4 ml of phage buffer. Resuspension was loaded on CsCl step gradient (1.2 g/ml, 1.4 g/ml, 1.6 g/ml) followed by ultracentrifugation 90,000 × *g* for 2 h at 4 °C. The phage band was extracted, dialyzed against phage buffer, and concentrated using a 30,000 MWCO concentrator. Subsequently, 10 μl was applied onto carbon-coated grids for 10 min, followed by three consecutive rounds of washing with 10 μl of distilled water, 30 sec each, and stained with 2% uranyl acetate for 1 min. Such prepared grids were analyzed with the electron microscope Tecnai T12 Spirit operating at 120 kV equipped with a LaB6 filament at the Center of Structural Biology, Imperial College London.

## Western blot analysis

For phi80 derivative with 3xflag-*taiEcA*(*orf63*) (C-terminal) and 3xflag-*orf6* (N-terminal), cultures were grown to $OD_{600}$ = 0.2 was reached and MC induced. For phi80 with 3xflag-*taiEcA* and 3xflag-*orf6* complemented with pBAD18 empty or Q gene, cultures were grown to $OD_{600}$ = 0.2 was reached and induced with 0.02% arabinose. Following phage induction or plasmid overexpression, cultures were incubated at 30 °C, and one milliliter of each sample was taken at defined time points and pelleted. Samples were frozen at − 20 °C until all time points were collected. Then, samples were re-suspended in 200 μL digestion/lysis buffer (50 mM Tris-HCl, 20 mM $MgCl_2$, 30% w/v raffinose) plus 5 μL of lysozyme (50 μg/mL), mixed briefly, and incubated at 37 °C for 1 h. Following incubation, 2x Laemmli sample buffer (Bio-Rad, 2-mercaptoethanol added) was added. Samples were heated at 95 °C for 10 min and incubated on ice for 5 min. Then, samples were run on 15% polyacrylamide gels and transferred to a PVDF transfer membrane (Thermo Scientific, 0.2 μM) using standard methods. Protein was detected using anti-Flag antibody probes (1:2,000 (v/v); Monoclonal ANTI-FLAG M2-Peroxidase (HRP) antibody (Sigma-Aldrich; A8592; clone M2, purified immunoglobulin, buffered aqueous glycerol solution; LOT# SLCF0816) as per the protocol supplied by the manufacturer.

## Identification of Tai immune systems and counter-defense

To identify more variants of the Tai immune system, we performed a PSI-BLAST, using TaiPectA and TaiPectB as a template and later other Tai versions, versus the NCBI database (with standard parameters, E value < 0.05). Those positive hits were manually analyzed to corroborate that they form part of the Tai immune system and compiled in Supplementary Table S3. In addition, the genomic origin of those positive hits was analyzed to discern whether they were encoded by a phage or phage satellites (P4-like elements or PICIs). A Similar strategy was employed to identify more counter-defense variants, performing a PSI-BLAST analysis with AtiEcA (Orf64) as a template, versus the NCBI database (with standard parameters, E value < 0.05).

## AlphaFold prediction

The structure models of proteins from different immune systems and the central tail fiber were generated using the AlphaFold Colab server. Predicted structures were compared using the structural multi-alignment server PROMALS3D.

## Bacterial adenylate cyclase-based two-hybrid (BACTH)

The BACTH assay for protein-protein interaction was conducted as previously described[68] using two compatible plasmids, pUT18 and pKT25, expressing the different protein combinations. Both plasmids were co-transformed into *E. coli* BTH101 and plated on LB supplemented with ampicillin, kanamycin, 0.1 mM of isopropyl-b-D thiogalactopyranoside (IPTG), and X-gal as an indicator. After incubation at 30 °C for 48 h, the protein-protein interaction was detected by a color change. Blue colonies represent an interaction between the two clones, while white/yellow colonies are negative for any interaction.

## Quantification and statistical analysis

For all quantitative data, three biological replicates were performed in accordance with standard practices in the field. For Southern blots, Western blots, and EM images, one or two replicates were performed. Sample sizes are indicated in the corresponding figure legends. Data are presented as mean ± standard deviation. All statistical analyses were performed as indicated in the figure legends using GraphPad Prism v.10.1.1 software. One-way ANOVA with Dunnett's multiple comparisons test was applied to compare three or more groups. An unpaired t-test was performed to compare the two groups. The statistical value for each experiment is listed in Supplementary Table S4.

## Reporting summary

Further information on research design is available in the Nature Portfolio Reporting Summary linked to this article.

## Data availability

The WGS data generated in this study have been deposited in the NCBI SRA database under accession code BioProject PRJNA1069670. We use ColabFold v1.5.5 AlphaFold2 using MMseqs2 server to obtain the trimeric structure of c-terminal J proteins of HK106 (residues 876-1183), phi80 (residues 913-1192) and HK97 (residues 876-1296) phages. The models have been deposited in Zenodo [https://doi.org/10.5281/zenodo.12783471]. Source data are provided in this paper.

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

## Acknowledgements
This work was supported by grants MR/X020223/1, MR/M003876/1, MR/V000772/1, and MR/S00940X/1 from the Medical Research Council (UK), BB/V002376/1 and BB/V009583/1 from the Biotechnology and Biological Sciences Research Council (BBSRC, UK), and EP/X026671/1 from the Engineering and Physical Sciences Research Council (EPSRC, UK) to J.R.P, a Royal Society Research Grant [RG\R2\232329] and an Imperial College Research Fellowship (ICRF) to A.F.-S., and a PID2022-136595NA-I00 from Spanish Government (Ministry of Science and Innovation) and a CDEIGENT Research Fellowship [CIDEIG/2022/19] from Valencian Government to L.M.-R.

## Author contributions
J.R.P. and A.F.-S. conceived the study; L.H., L.M.-R., J.B.P., N.A., and A.F.-S. conducted the experiments; L.H., L.M.-R., J.B.P., E.P.C.R., T.R.D.C., J.R.P., and A.F.-S. analyzed the data; J.R.P. and A.F.-S. wrote the manuscript.

## Competing interests
The authors declare no competing interests.
