## [Peer Review File · Nature Communications]

Tail assembly interference is a common strategy in bacterial antiviral defensesEditorial Note: This manuscript has been previously reviewed at another journal that is not operating a transparent peer review scheme. This document only contains reviewer comments and rebuttal letters for versions considered at *Nature Communications*.

Reviewer #2 (Remarks to the Author):

I think the MS has been improved in places but I still have reservations.

First, the response to point 1 is still unsatisfactory::

“1. I do not get the space devoted to community protection from phage degradation. It seems to me to be rather another selfish genome system, like the tailocins and bacteriocidal pili. Tai systems are just too specific to be general community protection systems like CRISPR and restriction.

We disagree with the reviewer on this point. Our experiments clearly demonstrate that the presence of these systems protects the population when the infecting phage is influenced by the Tai system. We have now provided a more detailed explanation of this in the discussion (lines 360-363).”

This is a strange distinction. If a phage infects a cell and phage tails are not assembled, then of course the surrounding population is protected. Fig. 3b shows that at i-moi (input MOI, to distinguish from actual MOI, which can only be determined from measuring adsorption) of 10, where presumably all host cells are infected, the cells all lyse (as expected since tail assembly is unrelated to lysis).at end of infection cycle. At lower iMOI, like 1 and 0.1, where there are many uninfected cells, the culture mass just keeps going. No phage produced so no lysis downstream. If phage cannot propagate, what is the point of doing all these curves and all these calculations? Fig 3bc can be deleted as uninformative.

It would be MUCH better to do traditional 1 step growth experiments to assess the defect caused by the Tai protein.. These plate reader OD assays are a poor substitute because of the variability in adsorption times. Moreover, if one looks closely at the uninfected plots, the cells are growing at a doubling time of 2 h – 4h or worse, throughout the experiment. Plate readers make for terrible growth conditions.

More importantly, the response to point 3 is still inadequate. I pointed out that TaiEcC seems to be able to inhibit podophage tail assembly (HK620, P22) and also myophage (p27 in Fig. 3A, P27 in the Table S1). This is a hugely important issue and would speak to the evolution of Tai proteins (i.e., how to evolve to attack other phage assembly steps). I note that no “escapees” were obtained for p27 on TaiEcC, so no candidate target can be assigned. I strongly recommend that the experiments in Fig. 4a and S5 be extended to TaiEcC inhibition of p27. Fig. 4A already shows that TaiEcC cannot block sipho tail assembly NS Fig. S5bc shows that it can block podo tail assembly but what is needed is to show that it can block myo (P27). I dont know why the authors didn’t jump on this after the first review. If TaiEcC can block both podo and myo, it greatly increases the punch line impact.

An alternative explanation is that a strain id mistake has been made. There is a P27 phage that is P22-like pdophage but it would be a Salmonella host. Fig. 2A says that “p27” is an E. coli host. So this is very confusing.

Minor

It would be good to add a column to Table S1 and specify myo, sipho, podo tail morphologies. The stupid Linnaean terminology has no value and can be deleted to save space.

It still says on 138ff “unrelated phages”....All the phages that are affected are lambdoid, so how is it possible to claim they are unrelated? In fact, it is strange that the phenomenon seems to be a lambdoid restricted. Did other temperate types like P2 never learn how to do it? The authors should comment on this.

Reviewer #3 (Remarks to the Author):

The revised manuscript is much improved, and in general, the authors have adequately responded to my comments.

One ‘major’ remaining issue pertains to my comment “the authors state that Tai proteins inhibit the tail assembly pathway, however, what is shown are EMs of intact capsids without tails. We cannot discern if tails are assembled but not attached or if they are not assembled at all. This is problematic given the language throughout the text; the authors invariably use the terms 'tail assembly' and 'tail formation'. Based on the data in the paper, there is evidence for a lack of tails attached to capsids, but whether tail components were not produced, the full tail was not assembled, or the tail was not attached to the capsid remains unclear.

From the methods, the EM images appear to be from phages post cesium fractionation – the other fractions should be analyzed for tails/tail proteins, etc”

In response, the authors performed an experiment, the results of which are shown in Figure S5A, claiming that ‘both capsids and tails were observed in the terL mutant phage, only a few tails attached to the capsids were observed in the sample obtained after the expression of the TaiEcA protein.’ However, the presented analysis is incomplete: the TEM of the terL mutant shows only tails (perhaps a single capsid); so while I am convinced they analyzed the fraction containing tails for the delta terL (and not the capsid fraction, despite Line 227) I am not convinced that analysis was done of ALL fractions for the phage during infection of the TaiEcA protein. The other fractions should be analyzed for tails/tail proteins and that data presented. In addition to EM, they can show the resulting fractionation (do the gradients look markedly different?) and do SDS PAGE analysis of the fractions if western blotting is not an option. I understand the authors are convinced of their own model, but based on the data presented and the degree to which they claim interference with assembly, more should be done to address this point.

Along these same lines, the EMs are not quantified, and representative images are shown; images should be quantified where possible to give an appreciation for the strength of the conclusions drawn.

Also, although the authors indicated my critique of the title, stating the tail assembly strategy is widespread, had been considered, and the title corrected, the title still includes the word widespread. The reported tai and anti-tai genes are from a small number of bacteria, where the majority of homologs presented are found in E.coli and S. enetrica. Hence, the use of the word widespread in the title is not justified.

Minor:

Line 332 end of the operon – should be beginning

Referees' comments:

Reviewer #2 (Remarks to the Author):

I think the MS has been improved in places but I still have reservations. First, the response to point 1 is still unsatisfactory: "1. I do not get the space devoted to community protection from phage deprecation. It seems to me to be rather another selfish genome system, like the tailocins and bacteriocidal pili. Tai systems are just too specific to be general community protection systems like CRISPR and restriction. We disagree with the reviewer on this point. Our experiments clearly demonstrate that the presence of these systems protects the population when the infecting phage is influenced by the Tai system. We have now provided a more detailed explanation of this in the discussion (lines 360-363)."

This is a strange distinction. If a phage infects a cell and phage tails are not assembled, then of course the surrounding population is protected. Fig. 3b shows that at i-moi (input MOI, to distinguish from actual MOI, which can only be determined from measuring adsorption) of 10, where presumably all host cells are infected, the cells all lyse (as expected since tail assembly is unrelated to lysis). At end of infection cycle. At lower iMOI, like 1 and 0.1, where there are many uninfected cells, the culture mass just keeps going. No phage is produced so no lysis downstream. If phage cannot propagate, what is the point of doing all these curves and all these calculations? Fig 3bc can be deleted as uninformative.

It would be MUCH better to do traditional 1 step growth experiments to assess the defect caused by the Tai protein. These plate reader OD assays are a poor substitute because of the variability in adsorption times. Moreover, if one looks closely at the uninfected plots, the cells are growing at a doubling time of 2 h – 4h or worse, throughout the experiment. Plate readers make for terrible growth conditions.

We thank the reviewer for their comments. We performed different iMOI infections to demonstrate that the TaiEcA can protect the bacterial community (especially for iMOIs of 0.1 and 1). Regardless of whether the community is formed by the same or multiple bacterial members, the community is protected as there will be less phages able to attack new members due to TaiEcA inhibiting the tail formation, and therefore, the formation of infective particles.

We agree with the reviewer that some graphs from Figure 3 are repetitive, therefore, we have moved some data to a supplementary figure. Additionally, as suggested, we performed the traditional one-step growth experiment for phage HK544 (new Fig. 3D) and the results obtained are similar to those observed using plate readers.

More importantly, the response to point 3 is still inadequate. I pointed out that TaiEcC seems to be able to inhibit podophage tail assembly (HK620, P22) and also myophage (p27 in Fig. 3A, P27 in the Table S1). This is a hugely important issue and would speak to the evolution of Tai proteins (i.e., how to evolve to attack other phage assembly steps). I note that no "escapees" were obtained for p27 on TaiEcC, so no candidate target can be assigned. I strongly recommend that the experiments in Fig. 4a and S5 be extended to TaiEcC inhibition of p27. Fig. 4A already shows that TaiEcC cannot block siphon tail assembly NS Fig. S5bc shows that it can block podo tail assembly but what is needed is to show that it can block myo (P27). I don't know why the authors didn't jump on this after the first review. If TaiEcC can block both podo and myo, it greatly increases the punch line impact.

We totally agree with the reviewer. Indeed, we tried two years ago to obtain the NS images for phage P27. However, we did not manage to observe the phage at that time. We would like to highlight that this phage is quite difficult to work with. To address this comment, we have established a new collaboration with Dr. Costa's group, who managed to solve the problem and obtain the EM images. In the new Figure 4, we can observe how TaiEcC blocks tail formation for P27. Additionally, we can observe that only "free tails" are present in the sample with pBAD18 empty, while phage tailless particles and no "free tails" are observed in the sample overexpressing the TaiEcC version. We are glad that the reviewer suggested again to perform this experiment and we were finally able to demonstrate our hypothesis.

An alternative explanation is that a strain id mistake has been made. There is a P27 phage that is P22-like pdophage but it would be a Salmonella host. Fig. 2A says that “p27” is an E. coli host. So this is very confusing.

See previous comment.

Minor

It would be good to add a column to Table S1 and specify myo, siph, podo tail morphologies. The stupid Linnaean terminology has no value and can be deleted to save space.

The Siphoviridae, Podoviridae and Myoviridae classification has been recently abolished, as suggested by reviewer #1. That is why we updated the nomenclature of phage families accordingly. We have added a new column in Table S1 with the Siphoviridae, Podoviridae and Myoviridae information.

It still says on 138ff “unrelated phages”....All the phages that are affected are lambdoid, so how is it possible to claim they are unrelated? In fact, it is strange that the phenomenon seems to be a lambdoid restricted. Did other temperate types like P2 never learn how to do it? The authors should comment on this.

We thank the reviewer for their comments. We refer to unrelated phages because the Tai systems can block phages with low DNA sequence homology (as mentioned now in line 95). These phages are unrelated in DNA sequence. To avoid any confusion, we have corrected the sentence.

Referee #3 (Remarks to the Author):

The revised manuscript is much improved, and in general, the authors have adequately responded to my comments.

One ‘major’ remaining issue pertains to my comment “the authors state that Tai proteins inhibit the tail assembly pathway, however, what is shown are EMs of intact capsids without tails. We cannot discern if tails are assembled but not attached or if they are not assembled at all. This is problematic given the language throughout the text; the authors invariably use the terms ‘tail assembly’ and ‘tail formation’. Based on the data in the paper, there is evidence for a lack of tails attached to capsids, but whether tail components were not produced, the full tail was not assembled, or the tail was not attached to the capsid remains unclear.

From the methods, the EM images appear to be from phages post cesium fractionation – the other fractions should be analyzed for tails/tail proteins, etc” In response, the authors performed an experiment, the results of which are shown in Figure S5A, claiming that ‘both capsids and tails were observed in the terL mutant phage, only a few tails attached to the capsids were observed in the sample obtained after the expression of the TaiEcA protein.’ However, the presented analysis is incomplete: the TEM of the terL mutant shows only tails (perhaps a single capsid); so while I am convinced they analyzed the fraction containing tails for the delta terL (and not the capsid fraction, despite Line 227) I am not convinced that analysis was done of ALL fractions for the phage during infection of the TaiEcA protein. The other fractions should be analyzed for tails/tail proteins and that data presented. In addition to EM, they can show the resulting fractionation (do the gradients look markedly different?) and do SDS PAGE analysis of the fractions if western blotting is not an option. I understand the authors are convinced of their own model, but based on the data presented and the degree to which they claim interference with assembly, more should be done to address this point.

We thank this reviewer for their support and criticism. We apologize for any confusion regarding Figure S5A. We have now explained on the material and methods section and in the figure legend that the EM images for Figure S5A are from phage precipitation only and not from post-cesium fractionation. Therefore, the whole sample was analyzed.

We would also like also to highlight that, as with the new analysis performed for the P27 phage (see previous comment and new Figure 4), we can observe some “free tails” when we analyze the phage in the presence of the empty plasmid, while we do not observe them when we overexpress the Tai

system. This observation corroborates our initial hypothesis that Tai proteins inhibit the tail assembly pathway.

Along these same lines, the EMs are not quantified, and representative images are shown; images should be quantified where possible to give an appreciation for the strength of the conclusions drawn.

We have quantified them and added the information on Table S4.

Also, although the authors indicated my critique of the title, stating the tail assembly strategy is widespread, had been considered, and the title corrected, the title still includes the word widespread. The reported tai and anti-tai genes are from a small number of bacteria, where the majority of homologs presented are found in E.coli and S. enetrica. Hence, the use of the word widespread in the title is not justified.

Corrected. We have now changed the title and removed the words novel and widespread. We have removed novel because in a parallel study, recently published, they identified a similar system present in *Pseudomonas* phages. We have included the word common in the title because of the identification of multiple systems that block tail formation.

Minor:

Line 332 end of the operon – should be beginning

Corrected.

Reviewer #2 (Remarks to the Author):

Edits are satisfactory.

Reviewer #3 (Remarks to the Author):

[No further comments for author]